# Phosphodiesterases in the Liver as Potential Therapeutic Targets of Cirrhotic Portal Hypertension

**DOI:** 10.3390/ijms21176223

**Published:** 2020-08-28

**Authors:** Wolfgang Kreisel, Denise Schaffner, Adhara Lazaro, Jonel Trebicka, Irmgard Merfort, Annette Schmitt-Graeff, Peter Deibert

**Affiliations:** 1Department of Medicine II, Gastroenterology, Hepatology, Endocrinology, and Infectious Diseases, Faculty of Medicine, Medical Center, University of Freiburg, 79106 Freiburg, Germany; 2Institute for Exercise and Occupational Medicine, Faculty of Medicine, Medical Center, University of Freiburg, 79106 Freiburg, Germany; descha@gmx.de (D.S.); adhara.lazaro@alumni.uni-heidelberg.de (A.L.); peter.deibert@uniklinik-freiburg.de (P.D.); 3Department of Pharmaceutical Biology and Biotechnology, University of Freiburg, 79104 Freiburg, Germany; irmgard.merfort@pharmazie.uni-freiburg.de; 4Department of Radiology–Medical Physics, Faculty of Medicine, Medical Center, University of Freiburg, 79106 Freiburg, Germany; 5Translational Hepatology, Department of Internal Medicine I, Goethe University Clinic Frankfurt, 60590 Frankfurt, Germany; jonel.trebicka@kgu.de; 6Faculty of Medicine, Medical Center, University of Freiburg, 79106 Freiburg, Germany; annette.schmitt-graeff@uniklinik-freiburg.de

**Keywords:** liver cirrhosis, portal hypertension, NO-cGMP pathway, sGC, PDE-5, cGMP, nitric oxide, hepatic zonation, sGC modulators, PDE-5 inhibitors

## Abstract

Liver cirrhosis is a frequent condition with high impact on patients’ life expectancy and health care systems. Cirrhotic portal hypertension (PH) gradually develops with deteriorating liver function and can lead to life-threatening complications. Other than an increase in intrahepatic flow resistance due to morphological remodeling of the organ, a functional dysregulation of the sinusoids, the smallest functional units of liver vasculature, plays a pivotal role. Vascular tone is primarily regulated by the nitric oxide-cyclic guanosine monophosphate (NO-cGMP) pathway, wherein soluble guanylate cyclase (sGC) and phosphodiesterase-5 (PDE-5) are key enzymes. Recent data showed characteristic alterations in the expression of these regulatory enzymes or metabolite levels in liver cirrhosis. Additionally, a disturbed zonation of the components of this pathway along the sinusoids was detected. This review describes current knowledge of the pathophysiology of PH with focus on the enzymes regulating cGMP availability, i.e., sGC and PDE-5. The results have primarily been obtained in animal models of liver cirrhosis. However, clinical and histochemical data suggest that the new biochemical model we propose can be applied to human liver cirrhosis. The role of PDE-5 as potential target for medical therapy of PH is discussed.

## 1. Introduction

The two cyclic nucleotides cyclic adenosine monophosphate (cAMP) and cyclic guanosine monophosphate (cGMP) are intracellular second messengers regulating important metabolic or regulatory pathways. They are formed by adenylate cyclases (ACs) or guanylate cyclases (GCs) and degraded by phosphodiesterases (PDEs). Thus, PDEs affect various metabolic processes, inflammatory mediator production and action or function of ion channels, muscle contraction, and myocardial contractility [1,2,3,4,5].

In general, 11 PDEs can be differentiated according to their structure and substrate specificity as to cAMP or cGMP [2]. Since every organ is endowed with an organ-specific set of PDEs, activation or inhibition of a PDE by a specific compound induces an organ-specific reaction.

The best known target of pharmacological modulation is PDE-5 [3,4]. An increasing number of PDE-5 inhibitors with different pharmacological profiles have been developed. Known indications are erectile dysfunction, pulmonal arterial hypertension [5], high altitude edema [6], and lower urinary tract syndromes [7]. Potential emerging applications [8] include heart failure [9], stroke, neurodegenerative diseases [10], diabetic nephropathy, peripheral arterial disease, peripheral neuropathy, intestinal motility disorders [7,11], COVID-19 (adjunct therapy only) [12], and—as we propose—cirrhotic portal hypertension (PH).

PH often occurs in patients suffering from liver cirrhosis. It is defined as an increase in portal venous pressure (PVP), which is partly caused by disturbed regulation of sinusoidal tone. The latter, in turn, is controlled primarily by the NO-cGMP pathway. Thus, the aim of this review is to describe current knowledge of the pathophysiology of portal hypertension with focus on soluble guanylate (sGC) and PDE-5, enzymes whose activities finally regulate cGMP availability. The role of PDE-5 as potential target for medical therapy of PH is discussed.

## 2. Pathophysiology of Portal Hypertension

The hepatic circulation is unique due to a dual blood supply from the portal vein and the hepatic artery [13,14,15,16]. The liver receives approximately 25% of total cardiac output, which is 800–1200 mL/min portal blood flow. The portal vein collects partially deoxygenated blood from intestinal organs (stomach, intestines, pancreas, spleen, gallbladder, and visceral fat) containing resorbed nutrients and other molecules and delivers 75–80% of total liver blood supply. The remaining well-oxygenated 20–25% of hepatic blood supply originates from the hepatic artery [16,17].

The development of liver cirrhosis is associated with structure-related alterations and various functional dysregulations [13,18,19,20,21]. As a consequence, alterations in the hepatic circulation and peripheral systemic circulation occur. Thereby, portal hypertension, defined as an increase in PVP is one of the earliest and most serious sequelae of liver cirrhosis and may lead to life-threatening complications such as esophageal variceal bleeding. Clinically, PH is classified as PVP (equivalent: as hepato-venous pressure gradient = HVPG) greater than 5 mmHg. It becomes clinically significant when it exceeds 10–12 mmHg [22,23,24].

The restriction in portal blood flow can occur in different anatomical locations. PH is classified as prehepatic (e.g., portal vein thrombosis), intrahepatic, and posthepatic (e.g., liver vein thrombosis) [18,25,26]. The intrahepatic form can further be differentiated into presinusoidal, sinusoidal, and postsinusoidal. Cirrhotic PH (sinusoidal PH) occurs most frequently and will be focused on in this review. Liver cirrhosis can be caused by a variety of diseases, toxic damage (e.g., alcohol abuse), infections (e.g., chronic hepatitis B or C), autoimmune diseases, and metabolic diseases. It evolves gradually from mild abnormalities to life-threatening complications like esophageal variceal bleeding, ascites, hepatic encephalopathy, dysfunction of the kidneys (hepato-renal syndrome) or the lungs (hepato-pulmonary syndrome, porto-pulmonary hypertension).

Clinical studies defined values for PVP that have to be achieved for a successful therapy: decrease of HVPG by ≥20% of baseline or to ≤12 mmHg [22,27,28,29,30]. By achieving these thresholds, potentially life-threatening complications of PH, such as variceal bleeding or rebleeding, spontaneous bacterial peritonitis, or dysfunction of other organs, may be prevented [22,27]. Additionally, a short term HVPG response to acute intravenous drug application (particularly propranolol) by ≥10% of baseline or a reduction to ≤12 mmHG [22,31] was found to predict a beneficial therapeutic effect in terms of clinical endpoints.

### 2.1. Elevated Intrahepatic Vascular Resistance

An increase in intrahepatic vascular resistance to portal blood flow is the main causative factor for the development of intrahepatic PH. There are two components involved. Firstly, there is a structural component which is related to the disturbed liver architecture, development of fibrosis, formation of regenerative nodules, angiogenesis, and vascular occlusion. It contributes about 70% to the intrahepatic vascular resistance. Secondly a dynamic component which accounts for the remaining 30% is involved. It is attributed to an increased sinusoidal tone caused by an impaired interplay of the sinusoidal cells leading finally to transformation of contractile hepatic stellate cells to myofibroblasts and a disturbed regulation of the NO-cGMP pathway [19,25,32,33].

Moreover, an elevated splanchnic blood flow due to dilation of splanchnic blood vessels is contributing to the development and/or worsening of PH [19,34].

The vascular and sinusoidal tone is mainly controlled by the NO-cGMP pathway (Figure 1A). NO is generated from L-arginine by endothelial NO-synthase (eNOS). NO diffuses to contractile cells and activates soluble guanylate cyclase (sGC) catalyzing the conversion of guanosine triphosphate (GTP) to cyclic guanosine monophosphate (cGMP). cGMP activates protein kinases (e.g., protein kinase G), cGMP-hydrolyzing phosphodiesterases (PDEs, particularly PDE-5), and cGMP-gated ion channels. It induces a decrease of intracellular calcium ions which finally leads to vascular dilation. cGMP is converted to functionally inactive 5′-GMP by PDE-5 [35,36,37,38,39,40,41,42,43,44,45,46,47,48,49,50,51,52,53,54]. PDE-5- inhibitors counteract the inactivation of cGMP.

Thus, structure-related alterations and functional dysregulations play a crucial role in pathogenesis of PH [13,18,19,20,21,55]. Since NO is a potent vasodilator and an essential component of the NO-cGMP pathway, research in the field of PH for years has been focusing on NO. The theory of the so-called “NO-paradox” was widely accepted: intrahepatic circulation is marked by an NO deficiency, whereas in the peripheral systemic circulation an NO excess exists [14,18,19,20,21,55,56,57]. In other words, intrahepatic vasculature is marked by vasoconstriction, while extrahepatic vasculature is characterized by dilation.

In this review we extend this view on the NO-cGMP pathway in liver cirrhosis with more current data suggesting a focus-shift from NO availability to cGMP availability.

We suggest the consideration of “cGMP-paradox” (low cGMP levels in sinusoids and high cGMP levels in the systemic peripheral circulation) as a more encompassing term to describe the hemodynamic disturbances in liver cirrhosis.

### 2.2. Intrahepatic Circulation

#### 2.2.1. The Hepatic Microvascular System

The hepatic microvascular system consists of hepatic arterioles, portal venules, sinusoids, and central venules. These are arranged into functional units [14,19] which allow the intermixing of arterial and portal blood for efficient extraction of oxygen and provision of nutrients to the parenchymal cells. These are lined by fenestrated sinusoidal endothelial cells (SECs) which lack a typical basement membrane [59] facilitating the high exchange capacity between sinusoidal blood and the space of Dissé and hepatocytes. Hepatic stellate cells (HSCs) are contractile cells that store Vitamin A and synthesize collagen and regulate the sinusoidal tone [15,33,60]. Kupffer cells (KC) represent hepatic macrophages.

#### 2.2.2. Sinusoidal Endothelial Cell Dysfunction

All sinusoidal cells are involved in the development of alterations leading to liver fibrosis/cirrhosis and PH. However, morphological and functional changes of SECs and HSCs are in the center of pathogenesis [14,18,19].

In liver cirrhosis, SECs loose one of their typical features—the “fenestrae”—and a basement membrane matrix is deposited on the cells, both impeding molecule exchange with liver parenchymal cells. This process is known as capillarization. The vascular endothelial growth factor (VEGF) is essential for maintenance of the fenestrae and for triggering NO release through eNOS in SECs and for maintaining HSCs in a quiescent state. NO stimulates soluble guanylate cyclase (sGC) in HSCs. This enzyme converts GTP to cGMP thus stimulating protein kinase G (PKG), which phosphorylates several proteins (including eNOS) and eventually induces dilation of the sinusoids [38,61,62,63]. A disruption of VEGF action abrogates this equilibrium leading to decreased cGMP synthesis and activation of HSCs. An interaction of eNOS with other proteins such as calmodulin, caveolin-1, HSP90, and Akt (protein kinase B) may play an additional role, as well as intracellular levels of the cofactor tetrahydrobiopterin and superoxide radicals [37,61,64].

SECs morphology is also influenced by mechanical shear stress which has a significant effect on eNOS, thus modulating perfusion and vascular tone within the sinusoids [20,57,65]. Apart from reduced vasodilating factors in cirrhotic liver an excess of constricting factors, such as endothelins, thromboxane A2, norepinephrin, angiotensin II, vasopressin, and leukotriens, exist [25,33,66,67]. However, the effect of these substances has less well been elucidated, particularly data on the effect of endothelin receptor antagonists are conflicting [68,69,70,71,72].

### 2.3. Activation of Hepatic Stellate Cells

The perisinusoidal HSCs are situated around the SECs and form finger-like extensions of their cytoplasm. Due to their contractile elements they are capable of regulating the sinusoidal diameter. This function is closely regulated by the NO-cGMP pathway as outlined earlier. In damaged liver, HSCs are activated, lose their capacity to store vitamin A, synthesize stress fibers (actin bundles), and produce extracellular matrix proteins and are transformed into myofibroblasts, contributing to sinusoidal contraction [20,32,73,74,75]. HSC activation is a complex process that involves multiple pathways and mediators and requires extracellular signals from resident and inflammatory cells [76].

Moreover, the activated HSCs display a diminished response to NO (less dilation) and an enhanced response to vasoconstrictors such as endothelin-1 (increased constriction) [68,69,70,71,72] and angiotensin II and through JAK2/RhoA/Rho-kinase activation to exaggerated contraction [77,78,79]. Therefore, both factors intensify the intrahepatic resistance via activated HSCs in liver cirrhosis.

#### 2.3.1. eNOS and iNOS

eNOS is constitutively expressed and uniformly distributed among the hepatic lobules in healthy liver. eNOS-derived NO attenuates HSCs activity and has a protective role in liver function. Activity of eNOS is reduced in cirrhosis contributing to decreased sinusoidal dilation [56,63]. The amount of eNOS is not lowered, but the translational modification through AKT or interactions with caveolin prevent full functional capacity [80]. However, other preclinical studies showed a reduced expression of eNOS in SECs, which is consistent with low NO formation and low activation of sGC [71,81,82,83,84,85]. Correspondingly, activation of eNOS and inhibition of RhoA/Rho-kinase by atorvastatin led to a decrease in PVP [84].

Leung et al. found an overexpression of inducible NO synthase (iNOS) in the CCl_4_-induced model of liver cirrhosis, but a decreased expression of eNOS [83]. Interestingly, several data showed this counteracting expression of the two NO-generating pathways [63,86,87]. Whereas eNOS-derived NO (in SECs) is pivotal to maintain physiological regulation of sinusoidal tone, maintain HSCs in a quiescent state, and is reduced in liver cirrhosis, iNOS is upregulated in response to pathological conditions, such as endotoxin or bacterial infections and can be found in all liver cells [81,88,89,90,91,92]. The detrimental effects induced by iNOS-generated NO (probably mediated by peroxynitrite) are cGMP-independent, whereas cytoprotective effects are mediated by cGMP [87]. Under physiological conditions, iNOS expression is minimal or even absent [63,86,93]. A possible but rather mechanistic explanation for this inverse regulation of the NOSs is the fact that enzymes compete for the same cofactor BH4 [71,88,89]. Therefore, iNOS upregulation could lead to reduced eNOS activity [94,95].

In contrast to several other studies, Schwabl et al. reported an upregulation of eNOS in bile duct ligation (BDL)-induced liver cirrhosis in progressed stages (not in early stages) and less moensin phosphorylation and myosin expression [96]: there was an increased expression of VEGF Receptor 2 (VEGFR2) and platelet derived growth factor β (PDGFβ) both in early and in advanced stages. In CCl_4_-induced liver damage eNOS remained constant. Schaffner et al. detected a significant 2.2-fold higher expression of eNOS mRNA in the thioacetamide (TAA)-induced model of liver cirrhosis [88]. iNOS could not be found in healthy liver, but a marked expression was observed in early and progressed stages of TAA-induced liver damage. A subsequent study of Uschner et al. using the BDL-model and the CCl_4_-model of liver cirrhosis confirmed the overexpression of eNOS and iNOS [97]. The latter data confirm the hypothesis of an iNOS upregulation in response to toxic stimuli. In addition, results of both studies suggest that an eNOS downregulation may not be a general feature of liver cirrhosis; instead it may depend on the stage of liver damage and the underlying etiology.

#### 2.3.2. sGC and PDE-5

By immunohistochemistry Theilig et al. detected sGC in nearly all HSCs of the periphery of the hepatic lobule in healthy liver [43]. The intensity decreased towards the central vein where nearly no sCG was detectable. These data suggest a zonation of sGC.

In the model of BDL-induced liver cirrhosis, Davies et al. demonstrated a decreased sGC activity, which was again increased by the addition of an NO-donor [98]. A zonation of sGC was not considered. Therefore, the lower sGC activity may indicate the loss of zonation in cirrhosis. cGMP levels were not measured. However, the authors suggested that addition of substrates for eNOS and application of a PDE-5 inhibitor might be beneficial on vascular dysfunction in liver cirrhosis.

Using Western blotting, Loureiro-Silva et al. observed a 1.6-fold overexpression of the β1-subunit of sGC in the model of CCl4-induced cirrhosis [99]. Likewise, the authors demonstrated an overexpression of PDE-5. A cellular mapping of the enzyme was not possible in this study, and no data were available about a potential zonation within the liver lobule.

Moreover, Lee et al. found a markedly increased protein expression of PDE-5 and a slight overexpression of sGC_α1β1_ in the BDL-induced liver cirrhosis [100]. One-week administration of sildenafil induced a further upregulation of sGC and a reduction of PDE-5. This effect was accompanied by a reduction of portal venous pressure and portal perfusion pressure.

In the study from Schwabl et al. the β1-subunit of the dimeric enzyme sGC was upregulated in BDL-induced liver cirrhosis and to a much lesser extent in the CCl_4_-induced liver cirrhosis [96]. In healthy liver, sGC was primarily detected in HSCs and portal venules, whereas only minor PDE-5 expression was observed. After bile duct ligation, sGC expression was shown in HSCs, hepatocytes, and Kupffer cells. This may indirectly reflect an altered distribution of enzymes of the NO-cGMP pathway in liver cirrhosis and will be discussed later. Further downstream factors (e.g., PDE-5, cGMP) were not investigated. According to Schaffner et al. both sGC subunits (α1 and β1) were overexpressed in the thioacetamide induced model of liver cirrhosis [88]. In addition, a marked overexpression of PDE-5 was detected, which most likely explains the low intrahepatic cGMP levels measured. Administration of the PDE-5-inhibitor sildenafil led to a normalization of cGMP levels and a lowering of PVP. Therefore, this paper yielded the molecular basis for the action of PDE-5 inhibitors on PVP in liver cirrhosis. Again, the study of Uschner et al. confirmed the overexpression of sGC and PDE-5 in the BDL-model and the CCl_4_-model of liver cirrhosis [97].

As shown by Hall et al. praliciguat stimulates sGC in the presence of the heme cofactor in a rat model of non-alcoholic steatohepatitis (NASH) [101]. sGC was localized in stellate cells or stellate-derived myofibroblasts, but not in hepatocytes. These results could be confirmed in human hepatocytes. However, sGC was detected in Kupffer cells and vascular smooth muscle cells, as well as in endothelial cells in small amounts. Praliciguat inhibited the transforming growth factor β (TGFβ)-mediated transformation of HSCs to myofibroblasts. Interestingly, both sGC subunits were determined in increased amounts in fibrotic tissue of the CCl_4_-induced liver damage, both on the protein and the mRNA level (about 2-fold). Gene expression of further components of the NO–sGC–cGMP pathway, such as PKG and vasodilator stimulated phosphoprotein (VASP), was higher in fibrotic than in healthy liver. Correspondingly, the increase of intrahepatic cGMP induced by praliciguat was higher in fibrotic compared to healthy liver. Likewise, in healthy human livers sCG_β1_ was immunostained in perisinusoidal cells (most likely stellate cells). In human NASH livers, there were unchanged hepatic regions with sGC_β1_ localization similar to healthy tissue. However, within the fibrotic bridges multiple sGC_β1_-high-positive clusters of fibroblast-like cells that co-localized with α-SMA were observed.

The paper of Hall et al. contains the most comprehensive data of alterations in the components of the NO-cGMP pathway in experimental liver cirrhosis [101]. However, a possible zonation of the respective enzymes and the potential role of PDE-5 were not taken into account. Regarding the sum of all currently available biochemical data for sGC and PDE-5, we suggest a model that shows the mechanism how PDE-5 inhibitors may lower portal pressure in cirrhotic portal hypertension (Figure 1B,C).

#### 2.3.3. Zonation of the Components of the NO–cGMP Pathway: Opposing Zonation of sGC and PDE-5

Evidently, there are contradicting results about the underlying causes of the dysfunctional NO-cGMP signaling in liver cirrhosis. Yet, current data from our own group might clarify some seemingly inconsistent results: Schaffner et al. [88] detected an increased expression of eNOS, and of both sGC subunits, but a marked overexpression of PDE-5. In cirrhotic animals, serum cGMP was reduced which could be nearly normalized by sildenafil, a PDE-5 inhibitor. In healthy rat livers a clear zonation of PDE-5 was found: low concentrations of PDE-5 in perisinusoidal cells (most probably stellate cells) along the sinusoids and a sharp increase of immunostaining in zone 3 hepatocytes. Combining these results with previous data on sGC zonation found by Theilig et al. [44] (sGC high in the periphery of the lobules, low around the central vein), it is likely that in healthy livers a high cGMP production in the peripheral parts of the hepatic lobule takes place, whereby cGMP may exert its physiological function inside the sinusoids. However, cGMP is degraded by high PDE-5 before entering the extrahepatic vasculature. In contrast, in cirrhotic livers immunostaining showed a marked PDE-5 overexpression and a loss of physiological hepatic zonation. In addition, the data of Schaffner et al. [88] with high PDE-5 immunostaining in fibrotic liver tissue find its counterpart in the sGC overexpression in fibrotic tissue as shown by Hall [101].

To illustrate the expression of PDE-5 in healthy and diseased rat and human liver tissue, tissue samples from an animal study and clinical studies were immunostained (Figure 2, Figure 3, Figure 4 and Figure 5). The animal research protocol was approved by the local institutional animal care and use committee (Regierungspräsidium Freiburg, Germany, ref. No. G-13/89). Animal care was performed in accordance to the rules of the German animal protection law and the animal care guidelines of the European community (2010/63/EU). The analysis of human tissue samples was part of a study that has been approved by the local ethics committee (Albert-Ludwigs-University, Freiburg, Germany, HBUF 474/14 and 299/01). All patients had signed informed consent.

In healthy rat liver tissue PDE-5 is expressed in perisinusoidal cells and is evenly distributed along the sinusoids (Figure 2). Moreover, there is a marked overexpression of the enzyme in zone 3 hepatocytes. In cirrhotic rat liver tissue, this zonation gets lost and an upregulation of PDE-5 in the deformed liver lobules can be found (Figure 3). Preliminary data show similar findings in human liver: in healthy human liver, PDE-5 immunostaining shows a roughly even distribution of the enzyme in perisinusoidal cells and a high PDE-5 immunostaining in zone 3 hepatocytes (Figure 4). In cirrhotic human liver the same pattern was detected as in rat cirrhotic liver (Figure 5).

The interdependency of sCG and PDE-5 expression for regulation of sinusoidal tone in healthy and diseased liver might represent an interesting new approach (Figure 6A,B). However, further data are needed to confirm its validity.

Considering the role of the disturbed hepatic zonation in pathophysiology of PH might be a new approach, although the concept of hepatic zoning itself is not new. It has already been exemplified by hepatic enzymes involved in different metabolic pathways [85,102,103,104,105,106,107,108,109]. Accordingly, correct hepatic zoning is apparently essential to ensure physiological metabolic liver functions. This may be also true for adequate NO and/or cGMP generation in response to prevailing conditions, which in turn is responsible for adequate regulation of PVP and sinusoidal tone. Liver cirrhosis is associated with loss of metabolic zonation due to structural modifications of the liver architecture and altered expression of many genes. The latter has extensively been investigated in the recent years using single cell RNA expressing techniques, both in hepatocytes and in non-parenchymal cells [110,111,112,113,114,115]. Alterations in expression, activity, and distribution of key enzymes and molecules, which are involved in the NO-cGMP pathway, as well as their specific role in PH pathophysiology have not been investigated in these studies. Recently, Su et al. [110] detected a disrupted zonation and increased expression of endothelin-receptor-B (ET-B) receptor in liver sinusoidal cells using single-cells transcriptomics. Reduced NO availability and increased constrictive response to endothelin-1 in liver cirrhosis could be a consequence of ET-B receptor disturbance. Apart from the sCG and PDE-5 zonation as described above this may be a further indication that zonation of components of the NO-cGMP pathway and its derangement in cirrhosis contributes to PH. The study of Ghallab et al. yielded further indications for a disturbed zonation of components of the NO–cGMP pathway [116]. It described the occurrence of “periportalization” in three different rat models of liver fibrosis. This refers to a shift in enzyme expression from periportal zones to pericentral hepatocytes, e.g., arginase, the enzyme that competes with eNOS for arginine, the substrate for NO formation, is highly expressed in pericentral zone 3 hepatocytes.

However, it has to be kept in mind that depending on the underlying etiology of liver damage (e.g., bile duct ligation or CCl_4_), different cells are activated, which might lead to variations in the disturbed zonation, activation of myofibroblasts, and fibrosis [76,117].

### 2.4. Further Vascular Alterations in Portal Hypertension

#### 2.4.1. Angiogenesis

Angiogenesis, the formation of new blood vessels, is a further important factor in both intrahepatic and peripheral systemic circulatory changes in PH. Its two forms (sprouting and splitting or intussusceptive angiogenesis) have been reported in physiologic conditions and in chronic liver diseases [118,119,120,121]. A characteristic feature of cirrhotic liver is the proliferation of blood vessels around regenerative nodules and fibrotic areas. Activated HSCs release angiogenic molecules such as angiopontin and VEGF. In turn, the SECs release PDGF and TGF-β thereby further facilitating HSCs migration and accumulation in vessels.

#### 2.4.2. Peripheral Systemic Circulation

The hyperdynamic circulatory state is a further important hallmark of PH. It is characterized by a decreased peripheral systemic vascular resistance, accompanied by an increased splanchnic blood flow thus aggravating the effect of increased intrahepatic resistance and perpetuates the elevated PVP, making it a vicious cycle [34,122,123,124]. The cardiac output and heart rate (HR) mostly are increased. However, due to peripheral systemic vasodilation the arterial blood pressure is mainly decreased [125].

This vasodilatation has been attributed to abnormal function in the endothelial cells, smooth muscle cells, and the adventitial layer containing neuronal termini. In the splanchnic vasculature there is an excessive NO production by eNOS among patients with PH [38,125,126,127]. In experimental liver cirrhosis it was shown that mild elevation of PVP might initiate the eNOS upregulation via VEGF [128,129,130]. Complex regulatory mechanisms are responsible for eNOS activation and subsequent NO overproduction. VEGF, hypoxia, shear stress, and inflammatory cytokines may activate the eNOS signaling pathway. Further factors may include an upregulation of GTP-cyclohydrolase I mediated by bacterial translocation [131], binding of eNOS to HSP90 [14,62,126], the Akt-dependent eNOS phosphorylation, among others [124]. The renin-angiotensin system could also play a role. Angiotensin II is generated via angiotensin converting enzyme (ACE) and is a peripheral vasoconstrictor. It is further degraded by ACE2 to the vasodilating peptide Ang (1–7), which binds to the G-protein coupled receptor Mas and leads to eNOS activation and consecutively to excess NO production [19,110,132,133].

Furthermore, several other molecules (e.g., carbon monoxide, prostacyclin, adrenomedullin, endocannabinoids, and endothelium-derived hyperpolarizing factors) may be involved in arterial splanchnic vasodilatation, suggesting the multifactorial nature of this process.

A decreased contractile response to vasocontractile molecules (e.g., neuropeptide Y, urotensin II, angiotensin, and bradykinin) is a further feature of splanchnic vasodilation. Such hypocontractility is partly caused by the excessive production of the vasodilators (endocannabinoids, adrenomedullin, calcitonin gene-related peptide (CGRP), atrial natriuretic factor (ANP), glucagon, carbon-monoxide, prostacyclin (PGI2)) which mostly act via eNOS mediated NO hyperproduction. In addition a diminished contractile response to α_1_-adrenerig agonists, angiotensin II, endothelin, and vasopressin was shown [124], as well as impaired contractile RhoA/Rho-kinase signaling in the smooth muscle cells.

However, the exact biochemical mechanisms have not been extensively studied compared to the regulation of the sinusoidal tone. Most studies identified rather an overexpression of eNOS than a suppression of this enzyme [134,135]. cGMP as the final dilating agent was found to be elevated both in aortal rings and in plasma of animals with cirrhosis [135]. The recent study of Uschner et al. also reported increased cGMP levels in peripheral blood vessels, which were most likely caused by decreased PDE-5 expression (detected in animals and humans with cirrhosis) [97].

In human liver cirrhosis plasma cGMP is significantly increased—particularly in patients with hepatic encephalopathy [136,137,138]—which might explain the peripheral systemic vasodilation [139,140]. It is questionable whether the observed changes of NO generation and sGC activities alone can explain these facts.

## 3. Targeting Components of the NO-cGMP Pathway for Therapy of Portal Hypertension

There is growing evidence that in animal and human liver cirrhosis characteristic alterations exist in the NO-cGMP pathway that are key components of the dynamic component of PH. As a logical consequence, availability of cGMP, the most important vasodilator, may be in the focus of future potential therapies for this life-threatening disease. Thus, NO, sGC, cGMP, and PDE-5 may be potential targets in PH therapy. Interestingly, PDE-5 overexpression is not a unique feature of liver cirrhosis, but can also be found in some tumors, in pulmonary arterial hypertension (PAH), and right heart hypertrophy [141,142,143,144].

### 3.1. NO

Intrahepatic NO availability can be increased by several measures which are partially addressed in Section 3.4.

### 3.2. sGC

The next potential target downstream of NO is the enzyme sGC. Two subsets of sGC modulators can be differentiated both leading to elevated intrahepatic cGMP generation. Stimulators of sGC, e.g., riociguat and IW-1973, act on the intact enzyme containing non-oxidized heme (Fe^2+^). They work synergistically with NO. In contrast, sGC activators, e.g., BAY 60-2770, act if the heme group is oxidized (Fe^3+^) or detached from the enzyme. They are capable of activating the enzyme when it does not respond to NO [145,146,147,148,149].

Data about hemodynamic changes induced by sGC modulators in the context of liver cirrhosis are still limited. Only one recent preclinical study exists. In this study the effect of chronic administration of the sGC stimulator riociguat (1 mg/kg, 1 × daily, intraperitoneal) was investigated over 2–3 weeks using the models of BDL- and CCl_4_-induced liver cirrhosis [96]. In the BDL-model, riociguat significantly decreased PVP by 24% in animals in an early and advanced disease stage, while mean arterial blood pressure (MAP) and HR were not affected. In the CCl_4_-model, riociguat significantly reduced PVP by 16% and tended to lower portosystemic shunting without deteriorating MAP in animals in an early disease stage. In contrast, riociguat exhibited no effects on PVP on MAP in animals in an advanced disease stage. Moreover, the effect of a chronic 1-week administration of riociguat was tested in a non-cirrhotic portal hypertensive partial portal vein ligation model. A significant reduction of PVP by 16% was observed accompanied by a significant decrease in MAP by 17% and significant increase of portosystemic shunting. The sGC stimulator IW-1973 had a protective effect on hepatic steatosis, inflammation, and fibrosis in a NASH-model [150,151]. The sGC activator BAY 60-2770 induced a reversal of capillarization and had antifibrotic effects in experimental liver damage [152,153]. The antifibrotic effects of sGC modulators in various fibrotic disorders were well described in a recent review by Sandner and Stasch [147].

### 3.3. PDE-5

Another potential target downstream of NO is the enzyme PDE-5, which can be inhibited by PDE-5 inhibitors, e.g., sildenafil, udenafil, vardenafil, and tadalafil. Sildenafil, the first specific inhibitor of PDE-5, was initially investigated as a potential drug to treat systemic arterial hypertension. However, the emergence of a blank side-effect directed the focus to its use as a new therapy of erectile dysfunction. The effect on systemic hypertension was only minor. Induction of dilation in penile arteries was initially recognized as the physiological background of sildenafil’s action in erectile dysfunction. Considering the action of PDE-5 inhibitors (reduction of cGMP inactivation) and knowledge about the pathophysiological background of sinusoidal constriction (low cGMP levels inside the cirrhotic liver) the application of PDE-5 inhibitors was suggested to treat PH.

One of the earliest preclinical studies by Colle et al. supposed that the PDE-5 inhibitor sildenafil might increase PVP in liver cirrhosis [154], but these data could never be confirmed. Halverscheid et al. investigated the hemodynamic effects of PDE-5 inhibitors in healthy rats in a preclinical study [155]. It was shown that after acute administration of sildenafil or vardenafil (1–100 µg/kg, intravenous) PVP remained unchanged or showed a trend towards a decrease. A dosage of 10 µg/kg, which was most effective with both PDE-5 inhibitors, led to a significant elevation in microvascular flow (corresponding to sinusoidal flow) by 15%. However, a significant reduction in mean arterial pressure (MAP) by 4% (sildenafil) and by 9% (vardenafil) was detected at the same time. Heart rate (HR) remained unaltered regardless of the applied dosage. Uschner et al. confirmed the effect of udenafil on PVP in the BDL- and CCl_4_-model of liver cirrhosis [97,156]. Their most recent study moreover investigated the acute effects of a combination of udenafil with the NSBB propranolol [97]. Administration of udenafil (1 mg/kg oral) reduced PVP by 25–35%, whereas the addition of propranolol led to a further lowering by 40%. Thus, a drug combination showed synergistic effects on PVP, whereas effects on systemic hemodynamics were minor. They reported that propranolol blunted the high cGMP levels in aortal tissue thus preventing increased vasodilation. Therefore, this study delivers the first rationale for a combination of PDE-5 inhibitors and NSBB.

Schaffner et al. evaluated the effects of acute administration of sildenafil (0.1–1 mg/kg, intravenous) in the model of TAA-induced liver disease [88]. A dose-dependent sildenafil effect was observed in animals in an early (fibrosis) and advanced stage (cirrhosis) of liver disease, whereas in healthy animals no clear-cut sildenafil effect was detected. The most noticeable change was observed after a high dose of sildenafil (1 mg/kg) in animals with cirrhosis. This led to a trend towards decreased PVP by 19%, a nonsignificant lowering of MAP by 17%, and a significant reduction of HR by 14%.

The BDL-model was also used in another study considering the effect of a chronic 1-week administration of sildenafil (0.25 mg/kg, 2 × daily, oral) [100]. Whereas in sham-operated rats no effect was found, a nonsignificant decrease in PVP and portal perfusion pressure, and a significant increase in microvascular flow were determined in diseased rats. These findings coincide with the results of a further study, which also used the model of BDL-induced liver disease [157]. Here, it was shown that PVP decreased by approximately 30% after chronic administration of the PDE-5 inhibitor udenafil (1, 5, or 25 mg/kg; 1 × daily, oral) for 3 weeks.

Chronic administration of PDE-5 inhibitors (udenafil or sildenafil) also exhibited antifibrotic effects in some first experimental studies—equivalent to the findings for sGC modulators [157,158]. Therefore, it might be possible that a prolonged chronic administration of sGC modulators or PDE-5 inhibitors could exert beneficial therapeutic effects.

In the first clinical study dealing with effects of PDE-5 inhibitors on HVPG and portal venous flow the acute effects of the PDE-5 inhibitor vardenafil were investigated [159]. Vardenafil (10 mg, oral) induced an increase of portal venous flow by 19% (*n* = 18) in healthy individuals and by 26% (*n* = 18) in patients with liver cirrhosis (Child A). HVPG was reduced by 19% (*n* = 5). Both effects suggested a marked reduction of intrahepatic blood flow resistance. In a case report including a patient with porto-pulmonary hypertension (combination of PVP and pulmonal arterial pressure) both, vardenafil and tadalafil, lowered pulmonal arterial pressure as well as PVP by 30% [160].

Subsequent studies partly supported these data, but contradictory results were also obtained. Lee et al. applied sildenafil (50 mg, oral) to seven patients with liver cirrhosis [161]. As expected, NO and cGMP in the hepatic veins increased and pulmonary vascular resistance and hepatic blood flow decreased. However, HVPG remained constant.

Clemmesen et al. tested sildenafil (50 mg, oral) in patients with liver cirrhosis [162]. Considering the total group, the decrease of HVPG from 18 to 16 mm Hg was not significant. However, in patients with less-progressed liver cirrhosis HVPG markedly decreased. These data may indicate that PDE-5 inhibitors exert their positive effects on PVP preferentially in early stages of liver damage, where the responsiveness of sinusoids is still preserved. Tandon et al. investigated the acute effect of 25 mg sildenafil in 12 patients with liver cirrhosis and did not observe any effect on PVP, but mean arterial blood pressure was decreased significantly [163]. Later data obtained from the study of Kreisel et al. suggested that the dose of 25 mg was too low to induce a relevant reduction of PVP [164]. In this study the effect of an acute and chronic 1-week administration of udenafil (12.5–100 mg, 1 × daily, oral) was tested in patients with compensated liver cirrhosis (Child A–B). A dosage of 75 mg or 100 mg was found to be most effective. After 1 h, HVPG was reduced by 25% (75 mg) or 17% (100 mg), respectively. Testing the acute effect again after the 1-week administration HVPG was lowered by 14% (75 mg) or 17% (100 mg), respectively. By combining the results of these two dosages a significant decrease in HVPG of 19% in the acute setting was found, while HR remained unchanged. However, the decreased HVPG was associated with a significant lowering of MAP of 4% in the acute setting and of 6% in the chronic setting which was clinically irrelevant. According to several studies a lowering of PVP in the acute setting by >10% may predict a beneficial long-term effect on clinical endpoints for PH [22].

There is only very limited data about effects of long-term use of PDE-5 inhibitors in PH. In the first case report of a male patient with porto-pulmonary hypertension vardenafil and tadalafil were reported to effectively lower pulmonary arterial and PVP [160]. However, after 1 year the patient was lost for a follow-up study.

Another current case report about a female patient with compensated liver cirrhosis (Child A) caused by primary biliary cirrhosis revealed promising results for the permanent use of PDE-5 inhibitors [165]. This patient has had several variceal bleedings and did not tolerate propranolol. In the acute setting, vardenafil (10 mg) led to a lowering of HVPG by 14%. This was accompanied by an increase of portal flow as verified by Doppler sonography and MRI. For the maintenance medication over the following 9 years with tadalafil (5 mg, 1 × daily, oral), similar effects on HVPG were reported. MAP also slightly decreased in the acute and in the chronic treatment phase. However, alterations were described to be clinically irrelevant. Interestingly, the biochemical liver function tests remained constant and no further variceal bleeding occurred. Table 1 gives an overview of previous studies on the effect of PDE-5 inhibitors on portal or systemic circulation.

### 3.4. PDE-5 Inhibitors as Therapeutic Alternative to Treat Portal Hypertension

According to the Baveno VI guidelines PH management involves pharmaceutical, endoscopic, and interventional therapies which are described elsewhere in more detail [22,27,166,167]. The application of nonselective beta blockers (NSBBs) is the current reference standard in pharmacological therapy of PH [22,27,168,169,170]. NSBBs act in two different ways: whereas a ß1 blockade reduces portal inflow by decreasing the heart rate, systemic arterial blood pressure, and cardiac output, a ß2 blockade leads to unopposed α1 activity resulting in splanchnic vasoconstriction thus reducing blood inflow towards the liver [166,167,171]. NSBBs may reduce the risk of hemorrhage from esophageal varices in patients at risk by 50% [27]. However, only about 30–40% of patients can achieve the required reduction of PVP due to intolerable side effects, such as bradycardia or systemic hypotension or increase of hepatic resistance [27]. In advanced stages of liver cirrhosis, complicated by hepatorenal failure, tension ascites, or in case of portopulmonary hypertension administration of NSBBs may have deleterious effects [172,173,174,175,176]. Further side effects are diminution of acral perfusion, depression, and erectile dysfunction.

The addition of organic nitrates to NSBBs may lead to a further reduction in PVP [177,178]. Hemodynamic response to this combined medical treatment is usually sustained after a long-term follow-up [179]. Results from studies using organic nitrates as monotherapy as primary prophylaxis were inconsistent [177]. Organic nitrates reduce the mean arterial pressure and exaggerate the hypotonic effects of beta blockers. In addition, the blood flow in collaterals is increased, at least in the short response, what may lead to bleeding episodes [180]. However, the combination of NSBBs and organic nitrates was not significantly better than NSBBs alone regarding overall bleeding or mortality rates, and had a higher rate of side effects (e.g., headache and lightheadedness) [181].

Other efforts were made to increase the availability of NO in the hepatic circulation. The intravenous application of L-arginine in patients with PH led to an increase in portal blood flow and a minor increase in cardiac output. In opposite to a decrease in mean arterial pressure the HVPG increased [182]. As an augmented NO production in the splanchnic vasculature is the major factor leading to splanchnic vasodilatation and exacerbation of PH [63,124,183], a further supply with NO-donors, which act systemically, is not reasonable. An NO-releasing derivative of ursodeoxycholic acid was used to avoid systemic and splanchnic side effects of organic nitrates [184,185]. However, in a clinical study the drug failed to reduce PVP but reduced systolic blood pressure and hepatic blood flow, suggesting that systemic effects predominate intrahepatic changes [186]. In order to selectively enhance intrahepatic NO availability further agents were studied. Statins (3-hydroxy-3-methylglutaryl-coenzyme A reductase inhibitors) induce an upregulation of NO production in the intrahepatic vasculature through an enhancement in endothelial NO synthase activity [187]. In animal studies as well as in clinical trials of short duration a reduction in HVPG by statins was observed, and the effects were additive with those of NSBBs [80,84,188,189]. The effect on rebleeding rate was only modest, but in some studies survival rate was increased and risk of decompensation was reduced [189,190,191,192]. Nevertheless, the potential risks of hepatotoxicity have to be kept in mind [193].

Other drugs, such as spironolactone, pentoxifylline, prazosin, molsidomine, prostanoids, endothelin receptor antagonists, and angiotensin II receptor agonists have been proposed as possible therapies, but these agents have not passed into clinical routine for prolonged therapy of PH. Newer reviews addressed the issue of novel vasoactive therapies [75,167,194,195,196].

Due to the pivotal role of the NO-cGMP pathway in the pathogenesis of PH the ideal therapy should specifically target intrahepatic vasculature and lead to a local enhancement of cGMP availability to counteract sinusoidal constriction. This could partly be achieved by application of sGC stimulators/activators and/or by PDE-5 inhibitors, both modulating cGMP availability. As outlined above both preclinical and clinical studies substantiated a positive effect of PDE-5 inhibitors in PH. A combination of sGC stimulators/activators and PDE-5 inhibitors has not been tested so far, but the combination seems to be reasonable. Moreover, as recently described by Uschner et al. [97], a combination of NSBBs and PDE-5 inhibitors may have synergistic effects thus avoiding side effects of NSBBs and increasing patients’ compliance. Further clinical studies are needed to confirm these hypotheses.

There is concern about the use of PDE-5 inhibitors in liver cirrhosis due to its potentially harmful effect on systemic blood pressure. If they are used for therapy of PH in the clinical setting their altered pharmacology, which are metabolized mainly by Cyt 450 3A4 or 2C9, in liver cirrhosis must be considered. Clinical studies exist only in patients with liver cirrhosis Child A and B, where it was shown that both AUC and C_max_ are altered. E.g., AUC of sildenafil increased by 85% and C_max_ increased by 47%, AUC of vardenafil AUC increased by 100–130% [197]. Administering the adequate dose of a PDE-5 inhibitor adjusted to the stage of liver cirrhosis the common adverse effects, such as hypotension, myalgia, back pain, headache, flushing, dyspepsia, rhinitis, and visual disturbances must be considered. PDE-5 inhibitors are contraindicated in patients with unstable angina pectoris, recent myocardial infarction, and poorly controlled hypertension. An interference with different drugs, particularly organic nitrates or α1-adrenoreceptor blockers, must be considered [198]. It should be investigated in clinical studies, whether a combination with NSBB—the current standard medical therapy of PH—is possible.

## 4. Conclusions

The current view of pathophysiology of PH has led to the hypothesis of the “NO-paradox” which describes a reduced NO availability inside the liver and an elevated NO production in the peripheral systemic circulation, necessitating a tailored therapeutic approach. However, reviewed data suggest that deranged cGMP availability better explains the contrasting findings of intrahepatic vasoconstriction and peripheral systemic vasodilation than the mere focus on NO. Thus, we suggest considering the term “cGMP-paradox” to describe the circulatory findings in liver cirrhosis: eNOS and sGC are overexpressed in cirrhosis. However, this effect is overridden by a very marked overexpression of PDE-5. This results in low levels of cGMP inside the cirrhotic liver leading to sinusoidal constriction. Inhibition of PDE-5 normalizes cGMP levels and lowers portal pressure. In peripheral arteries both eNOS and sGC are upregulated, however PDE-5 is downregulated. This results in high peripheral cGMP and systemic vasodilation.

Since altered activities and/or zonation of sGC and PDE-5 may play a pivotal role in this process, these enzymes should be investigated more systematically as potential targets in medical therapy of PH. Moreover, since there are first hints showing antifibrotic effects induced by sGC and PDE-5, these might represent interesting targets for the medical therapy of liver fibrosis/cirrhosis.

## Figures and Tables

**Figure 1 ijms-21-06223-f001:**
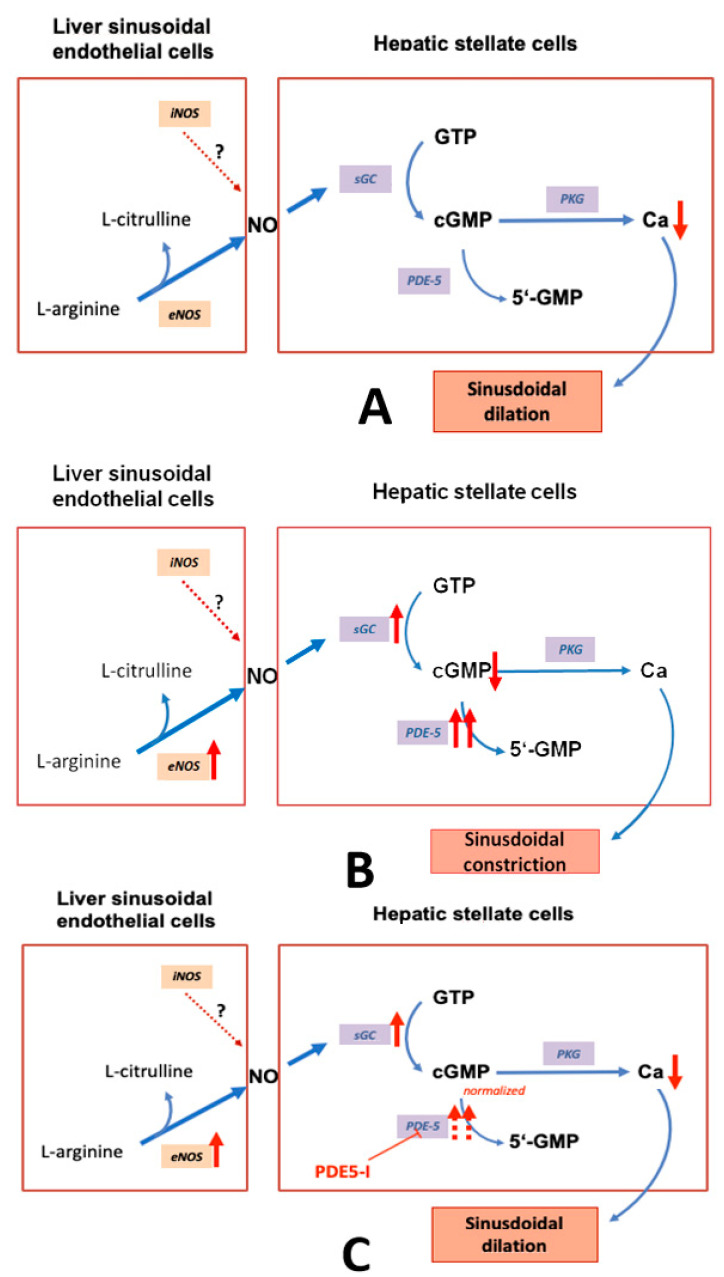
The nitric oxide-cyclic guanosine monophosphate (NO-cGMP) pathway, a regulator of sinusoidal tone (adapted from Schaffner [58]).(**A**):Regulation of sinusoidal tone in healthy livers: the activation of the NO-cGMP pathway takes place once NO is generated by eNOS in sinusoidal endothelial cells and diffuses into the neighboring hepatic stellate cells, where it binds to the enzyme sGC. The following activation of sGC, in turn, catalyzes the conversation of GTP to cGMP. cGMP, an intracellular second messenger, triggers distinct downstream signaling effects, which eventually exert vasodilation. As a negative feedback mechanism, rising cGMP concentrations initiate the activation of the enzyme PDE-5 which mediates cGMP inactivation. (**B**): Disturbed regulation of sinusoidal tone in liver cirrhosis: altered expression of key enzymes in the NO-cGMP pathway lead to reduced cGMP concentrations and thus sinusoidal constriction. (**C**): Effects of PDE-5 inhibitors in liver cirrhosis: application of PDE-5 inhibitors lead to a normalization of cGMP concentrations and thus sinusoidal dilation. (**↑** increased expression; **↑↑** markedly increased expression; **↓** decreased concentration; **⊥** inhibition).

**Figure 2 ijms-21-06223-f002:**
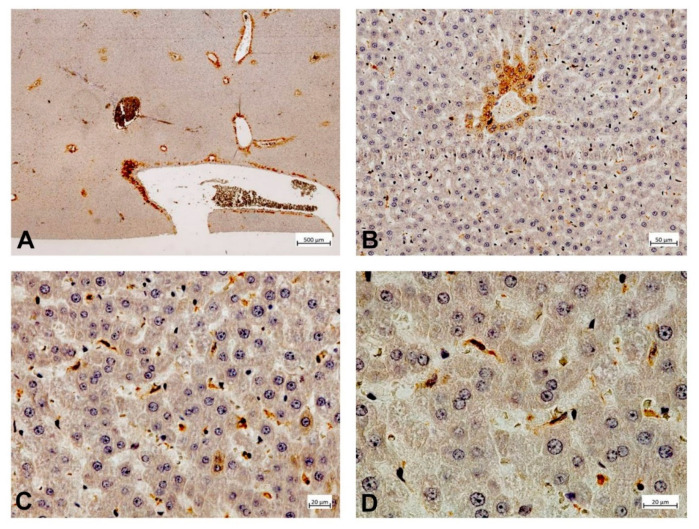
Phosphodiesterase-5 (PDE-5) expression in healthy rat liver tissue. Hepatocytes around the terminal hepatic venules (**A**,**B**) and adjacent to medium-sized intercalated veins (**A**) display strong cytoplasmic PDE-5 immunoreactivity. PDE-5 expression by sinusoidal lining cells is most prominent in zone 3 (**C**,**D**).

**Figure 3 ijms-21-06223-f003:**
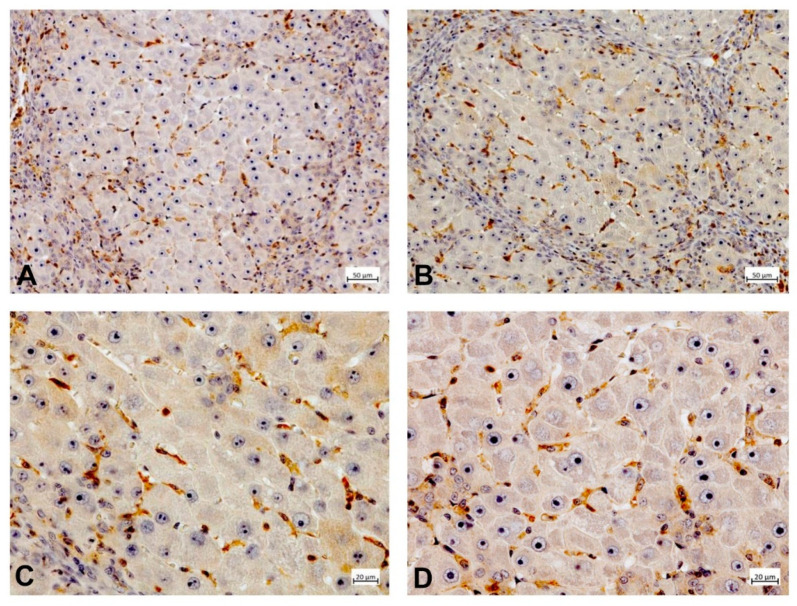
PDE-5 expression in cirrhotic rat liver tissue (thioacetamide (TAA)-model). Mesenchymal cells present in interconnecting fibrous septa (**A**,**B**) and sinusoidal lining cells of parenchymal islands (**A**–**D**) are immunolabeled by PDE-5 antibody. The architectural disturbance of liver parenchyma is associated with marked PDE-5 immunoreactivity of sinusoidal lining cells throughout the nodules (**C**,**D**).

**Figure 4 ijms-21-06223-f004:**
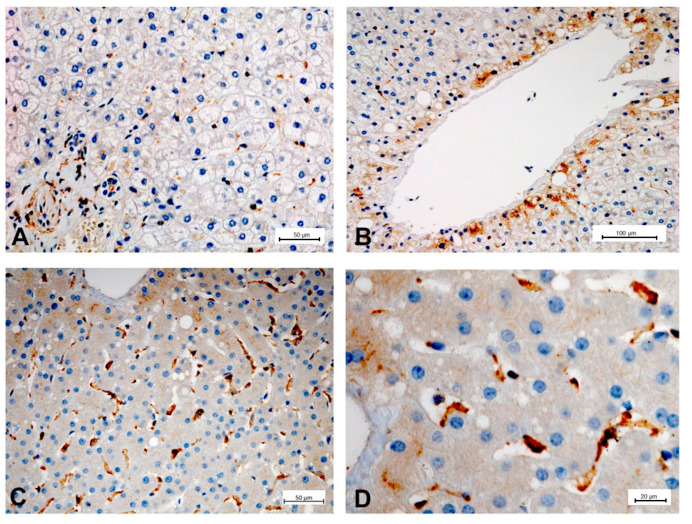
PDE-5 expression in healthy human liver tissue. Immunostaining for PDE-5 highlights portal tract vascular smooth muscle cells and myofibroblasts and shows reactivity of scattered sinusoidal lining cells in zone 1 and 2 (**A**). PDE-5 expression is enhanced in the cytoplasm of hepatocytes adjacent to terminal hepatic venules (**B**) and in zone 3 sinusoidal lining cells (**C**,**D**).

**Figure 5 ijms-21-06223-f005:**
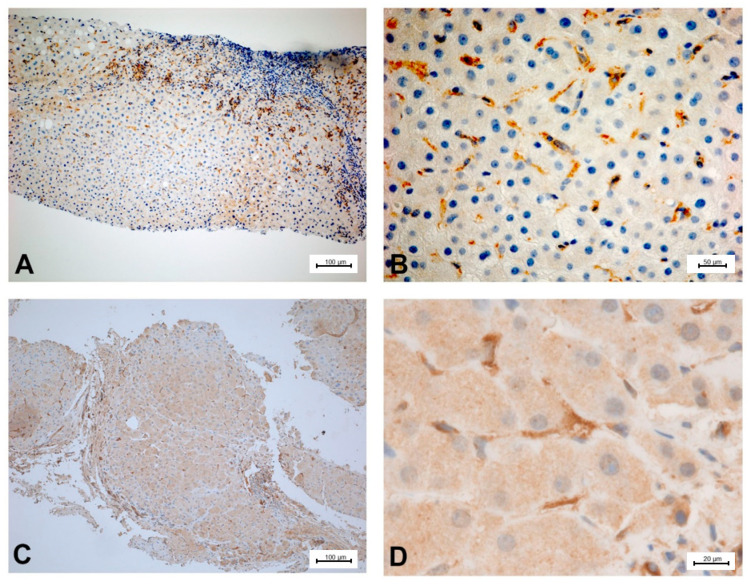
PDE 5 expression in cirrhotic human liver tissue. In advanced human liver fibrosis caused by chronic hepatitis B virus infection (**A**,**B**) and quiescent alcoholic micronodular cirrhosis (**C**,**D**) the nodular islands of hepatic parenchyma are surrounded by PDE-5-positive stromal cells predominantly present in fibrous septation (**A**,**C**). Note random distribution of PDE-5-expressing sinusoidal lining cells in parenchymal nodules (**B**,**C**).

**Figure 6 ijms-21-06223-f006:**
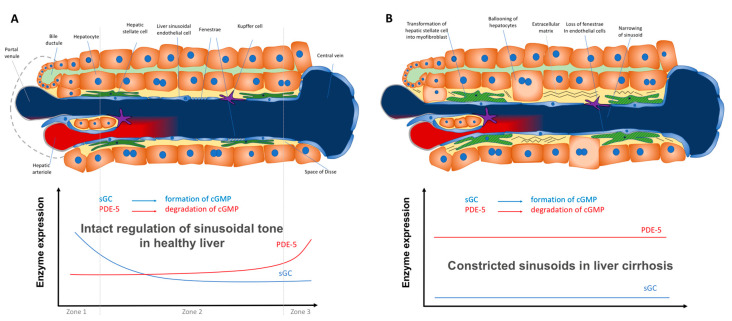
Regulation of sinusoidal tone by the key enzymes soluble guanylate (sGC) and PDE-5 in healthy and diseased liver. (**A**) In healthy livers an opposing zonation of sGC and PDE-5 may lead to a high cGMP production in the peripheral parts of the hepatic lobule, in which cGMP may exert its physiological function inside the sinusoids. However, excess cGMP might be degraded by high PDE-5 presence (zone 3) before entering the extrahepatic vasculature. (**B**) In cirrhotic livers the loss of physiological hepatic zonation, marked PDE-5 overexpression, and resulting increase in cGMP degradation might lead to sinusoidal constriction.

**Table 1 ijms-21-06223-t001:** Effects of PDE-5 on portal and systemic hemodynamics in animal and human experiments.

	Model	Compound	Dosage and Route	ΔMAP	ΔPVP	Remarks
Colle 2004 [154]	Wistar rats, BDL	Sildenafil	0.01–10 mg/kg, i.v.	−1–20% even more in sham rats	+2–+6% even more in sham rats	
Halverscheid 2009 [155]	Sprague Dawley rats, non-cirrhotic	Sildenafil Vardenafil	1, 10, or 100 µg/kg, i.v. 1, 10, or 100 µg/kg, i.v.	1.1; −3.9; −2.6% −11.0; −8.7; −7.4%	In all groups no increase, but decrease over time	3.3; 24.1; 18.3% 15.9; 29.2; 23.9% increase in portal flow
Schaffner 2018 [88]	Wistar rats, TAA	Sildenafil	0.1–1.0 mg/kg, i.v.	−14–−17%	−13–−19%	
Uschner 2020 [97]	Sprague Dawley rats, BDL or CCL_4_	Udenafil Udenafil/propranolol Udenafil 1 or 5 mg/kg	1 or 5 mg/kg 1 mg/kg	1 mg/kg: −20%; 5 mg/kg: −22% −7.5% 1 mg/kg: −31%; 5 mg/kg: −34%	−30–−23% −40% −30–−0%	
Schaffner 2018 [88]	Wistar rats, TAA	Sildenafil	0.1–1.0 mg/kg, i.v.	−14–−17%	−13–−19%	
Lee 2010 [100]	Sprague Dawley rats, BDL	Sildenafil, 1 week	0.25 mg/kg twice daily p.o.		−25%	
Choi 2009 [157]	Sprague Dawley rats, BDL	Udenafil for 3 weeks	1, 5, or 25 mg/kg p.o.		−14, −13, −31%	
Deibert 2006 [159]	Human, cirrhotic (*n* = 18)	Vardenafil	10 mg, p.o.		−19% (*n* = 5)	Hepatic arterial resistance and portal flow increased significantly
Bremer 2007 [160]	Human, cirrhotic PPHTN (*n* = 1)	Tadalafil	10 mg, p.o.		−30%	PAP −25%
Lee 2008 [161]	Human, cirrhotic (*n* = 7)	Sildenafil	50 mg, p.o.	Unchanged	+1%	MPAP and sinusoidal resistance significantly reduced
Clemmesen 2008 [162]	Human, cirrhotic (*n* = 10)	Sildenafil	50 mg, p.o.	−14%	−11%	
Tandon 2010 [163]	Human, cirrhotic (*n* = 12)	Sildenafil	25 mg, p.o.	−8%	−4% n.s.	
Kreisel 2015 [164]	Human cirrhotic (*n* = 30)	Udenafil	12.5; 25; 50; 75; 100 mg p.o. acute 6 days	Significant reduction with ≥75 mg in the acute setting and after 6 days	−3.5; −4.5; −7.5; −25.1; −17.3% −14.4; 3.1; −14.0; −13.5; −16.8%	
Deibert 2018 [165]	Human, cirrhotic (*n* = 1)	Vardenafil Tadalafil	10 mg 5 mg	−11%	−14% −15%	

BDL: bile duct ligation, MAP: mean arterial pressure; MPAP: mean pulmonary arterial pressure; PPHTN: portopulmonary hypertension; PVP: portal vein pressure; TAA: thioacetamide.

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
