# Peer review of "Phosphodiesterases in the Liver as Potential Therapeutic Targets of Cirrhotic Portal Hypertension"

_ijms, 2020, doi:10.3390/ijms21176223_

Round 1

Reviewer 1 Report

This is a review on the role of phosphodiesterase in the treatment of portal hypertension. The review is well written and structured but requires some minor revisions.

  1. Several references are missing for some statements that can be obvious but that require proper references. Ex: l.65 p2, l.66 p.2 l.72 p.2 definition of clinically significant portal hypertension, l144p.4., l.151 p.5 etc.
  2. l.141-142 p.4 Normal Liver sinusoidal endothelial cells do have a basement membrane that is tiny and not connected to LSEC and other cells but constituted of scattered collagens, fibronectin and proteoglycan (Müsch A. The unique polarity phenotype of hepatocytes. Exp Cell Res 2014) please correct .

  1. Please correct the following typos:

-Figure 1 legend, missing space between “tone.” And” the”

-l.93,p.3 scG.

  1. In the different sections, the authors are reviewing articles using experimental models of liver fibrosis in rodents. Variation of results are sometimes attributed to portal zonation. The authors have to be aware that beside zonation, bile duct ligation and CCL4 models probably activate different pool of cells leading to myofibroblasts and fibrosis. Indeed, it has been proposed, that BDL, activates quiescent periportal fibroblast and not HSC (Tsuchida T, Friedman SL. Mechanisms of hepatic stellate cell activation. Nat Rev Gastroenterol Hepatol 2017), Meier RPH, Interleukin-1 Receptor Antagonist Modulates Liver Inflammation and Fibrosis in Mice in a Model-Dependent Manner. Int J Mol Sci. 2019)

  1. A lexicon could be useful for the different enzymes

Author Response

Reviewer #1: Several references are missing for some statements that can be obvious but that require proper references. Ex: l.65 p2, l.66 p.

Answer: To clarify the statement in the text the following references were inserted

16. Eipel C, Abshagen K, Vollmar B. Regulation of hepatic blood flow: The hepatic arterial buffer response revisited. World J Gastroenterol 2010;16:6046–57 [PMID: 21182219 DOI: 10.3748/wjg.v16.i48.6046]

17. Kalra A, Yetiskul E, Wehrle CJ, Tuma F. Physiology, Liver [Internet]. In: StatPearls. Treasure Island (FL): StatPearls Publishing; 2020 [cited 2020 Aug 17]. Available from: http://www.ncbi.nlm.nih.gov/books/NBK535438  2 l.72 p.2 definition of clinically significant portal hypertension, l144p.4., l.151 p.5 etc.

Reviewer #2:2 l.72 p.2 definition of clinically significant portal hypertension, l144p.4., l.151 p.5 etc.

Answer: The following references were cited for definition of clinically significant portal hypertension 

22. de Franchis R, Baveno VI Faculty. Expanding consensus in portal hypertension: Report of the Baveno VI Consensus Workshop: Stratifying risk and individualizing care for portal hypertension. J Hepatol 2015;63:743–52 [PMID: 26047908 DOI: 10.1016/j.jhep.2015.05.022]

23. Jakab SS, Garcia-Tsao G. Evaluation and Management of Esophageal and Gastric Varices in Patients with Cirrhosis. Clin Liver Dis 2020;24:335–50 [PMID: 32620275 DOI: 10.1016/j.cld.2020.04.011]

24. Turco L, Garcia-Tsao G. Portal Hypertension: Pathogenesis and Diagnosis. Clin Liver Dis 2019;23:573–87 [PMID: 31563212 DOI: 10.1016/j.cld.2019.07.007]

Reviewer #3: 141-142 p.4 Normal Liver sinusoidal endothelial cells do have a basement membrane that is tiny and not connected to LSEC and other cells but constituted of scattered collagens, fibronectin and proteoglycan (Müsch A. The unique polarity phenotype of hepatocytes. Exp Cell Res 2014) please correct .

Answer: Müsch et al wrote in this paper: Of further importance, while most epithelia deposit a basal lamina, hepatocytes don't assemble extra cellular matrix (ECM)-molecules into a proper dense matrix, because laminin, an obligate basement membrane component and nidogen, a matrix crosslinker, are absent from the ECM surrounding mature hepatocytes.

Therefore, we modified the phrase to:

These (the sinusoids) are lined by fenestrated sinusoidal endothelial cells (SECs) which lack a typical basement membrane [59] facilitating the high exchange capacity between sinusoidal blood and the space of Dissé and hepatocytes. The paper of Müsch (#59) is cited.

Reviewer #4:Please correct the following typos:-Figure 1 legend, missing space between “tone.” And” the”-l.93,p.3 scG.

We corrected this typing error

Reviewer #5: In the different sections, the authors are reviewing articles using experimental models of liver fibrosis in rodents. Variation of results are sometimes attributed to portal zonation. The authors have to be aware that beside zonation, bile duct ligation and CCL4 models probably activate different pool of cells leading to myofibroblasts and fibrosis. Indeed, it has been proposed, that BDL, activates quiescent periportal fibroblast and not HSC (Tsuchida T, Friedman SL. Mechanisms of hepatic stellate cell activation. Nat Rev Gastroenterol Hepatol 2017), Meier RPH, Interleukin-1 Receptor Antagonist Modulates Liver Inflammation and Fibrosis in Mice in a Model-Dependent Manner. Int J Mol Sci. 2019).

Answer: Thank you for this information. We inserted the following phrase in p12, lines 403-405 and added these two references.

However, it has to be kept in mind that different types of liver damage activate different pools of cells leading to different types of disturbed zonation, activation of myofibroblasts and fibrosis [115,116].

On page 5 lines 216-217 we wrote.

HSC activation is a complex process that involves multiple pathways and mediators and requires extacellular signals from resident and inflammatory cells. We included the reference Tsuchida et al. (#76)

Reviewer #6: A lexicon could be useful for the different enzymes

Answer: It is a good suggestion from the biochemical point of view.However,we feel that this review article would not gain more clarity or readability for the clinician by adding a list of the enzymes (including their EC numbers).

Reviewer 2 Report

The authors reviewed the role of nitric oxide-cyclic guanosine monophosphate (NO-cGMP) pathway on the portal hypertension, focusing on two key enzymes, including soluble guanylatecyclase (sGC) and phosphodiesterase-5 (PDE5).

Overall, the manuscript was well-prepared and beneficial for readers. However, I have some comments to be addressed.

Comments:

1) They emphasized the potential utility of PDE-5 inhibitors all through the paper; however, the interest of physicians are not limited in the positive findings. Kindly mention the unfavorable /adverse events.

2) I would like to recommend to generate a summary table regarding the effects of PED-5 inhibitors to be easily understood by readers.

3) In the Figures 4 and 5, they used human samples. Kindly clarify the ethical appropriateness of the contents in the current paper (e.g, informed consent, approval of the ethical committee, etc).

4) The Figure 1 is blurred in comparison to the other figures. Kindly revise the quality of the figure.

Author Response

Reviewer #1: They emphasized the potential utility of PDE-5 inhibitors all through the paper; however, the interest of physicians are not limited in the positive findings. Kindly mention the unfavorable /adverse events.

Answer: We added the following remarks about potential risks or adverse events of PDE-5 inhibitors (page 17, lines 633-645)

There is concern about the use of PDE-5 inhibitors in liver cirrhosis due to its potentially harmful effect on systemic blood pressure. If they are used for therapy of PH in the clinical setting their altered pharmacology, which are metabolized mainly by Cyt 450 3A4 or 2C9, in liver cirrhosis must be considered. Clinical studies exist only in patients with liver cirrhosis Child A and B, where it was shown that both AUC and Cmax are altered. E.g. AUC of sildenafil increased by 85% and Cmax increased by 47%, AUC of  vardenafil AUC increased by 100-130%[197]. Administering the adequate dose of a PDE-5 inhibitor adjusted to the stage of liver cirrhosis the common adverse effects, such as hypotension, myalgia, back pain, headache, flushing, dyspepsia, rhinitis, and visual disturbances must be considered. PDE-5 inhibitors are contraindicated in patients with unstable angina pectoris, recent myocardial infarction, and poorly controlled hypertension. An interference with different drugs, particularly organic nitrates or α1-adrenoreceptor blockers, must be considered [198]. It should be investigated in clinical studies, whether a combination with NSBB – the current standard medical therapy of PH – is possible.

Reviewer #2: I would like to recommend to generate a summary table regarding the effects of PED-5 inhibitors to be easily understood by readers.

Answer: Very good suggestion. We generated a summary table about the effects of PDE-5 inhibitors in animal and human liver cirrhosis. Table 1. At the end of the text, after the References.

Reviewer #3: In the Figures 4 and 5, they used human samples. Kindly clarify the ethical appropriateness of the contents in the current paper (e.g, informed consent, approval of the ethical committee, etc).

Answer: We did so at the end of the text and included the votum of the ethics committeed for the animal study:

The analysis of human tissue samples was part of a study that has been approved by the local ethics committee (Albert-Ludwigs-University, Freiburg, Germany, HBUF 474/14 and 299/01). All patients had signed informed consent.

The animal research protocol was approved by the local institutional animal care and use committee (Regierungspräsidium Freiburg, Germany, ref. No. G-13/89). Animal care was performed in accordance to the rules of the German animal protection law and the animal care guidelines of the European community (2010/63/EU).

Reviewer #4: The Figure 1 is blurred in comparison to the other figures. Kindly revise the quality of the figure.

Answer: We replaced this Figure by a copy with better quality.

Reviewer 3 Report

Kreisel et al. reviewed that the PDEs in the liver is potential therapeutic targets of cirrhotic portal hypertension. Manuscript is written well.

  1. How was “non-cirrhotic portal hypertension”?
  2. In line 398, please make change to “1. NO”.
  3. The effects of LPS or hypoxia induced inflammation on PDE should be described.
  4. Pulmonary complications of portal hypertension should be added.

Author Response

Reviewer #1 and #2: How was “non-cirrhotic portal hypertension”?

Pulmonary complications of portal hypertension should be added.

Answer: We rewrote the paragraph about classification of portal hypertension as follows (Page 2, lines 78-86):

The restriction in portal blood flow can occur in different anatomical locations. PH is classified as prehepatic (e.g. portal vein thrombosis), intrahepatic, and posthepatic (e.g. liver vein thrombosis)  [18,25,26]. The intrahepatic form can further be differentiated into presinusoidal, sinusoidal, and postsinusoidal.  Cirrhotic PH (sinusoidal PH) occurs most frequently and will be focused in this review. Liver cirrhosis can be caused by a variety of diseases, toxic damage (e.g. alcohol abuse), infections (e.g. chronic hepatitis B or C), autoimmune diseases, and metabolic diseases. It evolves gradually from mild abnormalities to life-threatening complications like esophageal variceal bleeding, ascites, hepatic encephalopathy, dysfunction of the kidneys (hepato-renal syndrome) or the lungs (hepato-pulmonary syndrome, porto-pulmonary hypertension).

Reviewer #3: In line 398, please make change to “ NO”.

Answer: This was done.

Reviewer #4:The effects of LPS or hypoxia induced inflammation on PDE should be described.

Answer: It is a good suggestion to consider potential effects of LPS or hypoxia on PDE expression or function. A change in gene expression of both NOS and PDEs under special conditions (ischemia or hypoxemia, LPS, aging etc.) can be assumed, but has not yet been sufficiently investigated. We browsed the literature and found only few data, but mainly on a speculative level. However, data are available that vasoactive PDE inhibitors have positive effects in many conditions with underlying endothelial dysfunction, e.g. secondary to local or general hypoxia.

Round 2

Reviewer 2 Report

The authors revised their paper according to the comments. I consider the manuscript was well revised.